# Effect of formic acid treatment on *Apis mellifera* foraging behavior using nanopore metabarcoding technologies

Claudia L. Wiese[1]*, Rodolfo S. Probst[1,2,3], Heather M. Briggs[1], Joshua G. Steffen[1,2]

**1** Science Research Initiative (SRI), College of Science, University of Utah, Salt Lake City, Utah, United States of America, **2** School of Biological Sciences, University of Utah, Salt Lake City, Utah, United States of America, **3** Department of Entomology and Nematology, University of California, Davis, California, United States of America

* claudialwiese@gmail.com

## Abstract

The Western honeybee (*Apis mellifera*) is a crucial contributor to worldwide agriculture and ecological health but is experiencing wide population declines linked to *Varroa* mite infection. Formic Acid (FA) has been increasingly used to control for *Varroa*, yet its effects on *A. mellifera* hives, particularly their foraging preferences, remain unclear. In this study, we used a combination of pollen DNA metabarcoding with real-time nanopore sequencing to assess how FA treatment influences *A. mellifera* foraging preferences. DNA sequencing was performed on pollen samples from six University of Utah campus honeybee hives (n = 6) separated into FA and control treatments. Samples were collected before, during, and after FA application. We amplified *trnL,* a chloroplast DNA region useful for plant identification using portable sequencers from Oxford Nanopore Technologies (ONT). We detected a significant difference in foraging composition between FA-treated and placebo-treated hives at the end of the experiment. Control hives foraged from a more diverse array of plant genera. We also found individual hives to have unique foraging preferences independent of treatment. These findings suggest that FA treatment is associated with detectable differences in *A. mellifera* foraging behavior. The magnitude of FA impact, however, on hive foraging repertoire remains unclear. Pollen DNA metabarcoding with nanopore technology is an effective method for analyzing bee foraging patterns and holds significant potential for advancing ecological research on pollination health.

## Introduction

The western honeybee (*Apis mellifera*) is a crucial contributor to worldwide agriculture and ecological pollination [1,2]. *Apis mellifera* is estimated to contribute to €153 billion (~$170.5 billion) worth of crops and products every year [3] and pollinate up to a third of the United States food crops [4]. Honeybee presence in ecosystems across

**Data availability statement:** All broad reads and attached information can be found on the NCBI public repository under the identification labels of SubmissionID: SUB15836198 and BioProject ID: PRJNA1381099. R Scripts and appropriate files to run scripts have been added to DRYAD DOI: 10.5061/dryad.xksn02vw8.

**Funding:** The author(s) received no specific funding for this work.

**Competing interests:** The authors have declared that no competing interests exist.

the globe can be linked to food security, rural community income, and improvement of natural ecosystem functions [5,6]. Unfortunately, *A. mellifera* is suffering from widespread health decline and colony loss, deemed Colony Collapse Disorder (CCD). In the United States, 43% of hives were lost between April 2019 to April 2020 [7]. Europe and China report similarly destructive levels of hive losses [8]. Many factors contribute to this decline, such as habitat loss, exposure to pathogens, pesticides, and effects of anthropogenic climate change [9,10].

A leading documented cause of CCD is infection by *Varroa destructor* (hereafter Varroa), an ectoparasite [11,12]. These mites feed off larval hemolymph, damaging the subsequent development of workers and significantly decreasing their lifespan [13,14]. Additionally, and arguably more detrimentally, Varroa transmits viruses, such as Deformed Wing Virus (DWV) which can kill honeybees within one day of infection [15,16]. If left untreated against Varroa, *A. mellifera* colonies are likely to collapse within 2–3 years [17]. Currently, various treatment methods are used by beekeepers to prevent and control Varroa infections, with short-term application of Formic Acid (FA) into hives being one of them [18]. FA is the second most common chemical treatment against Varroa in Europe and is increasing in popularity within the United States [19,20]. FA is applied by spraying or by placing FA-soaked pads at the bottom of the hive (Fig 1). The strong efficacy of FA on lowering mite loads in hives is confirmed in many studies [21–23]. FA is hypothesized to affect the mitochondrial respiration of Varroa by inhibiting the enzyme which promotes the electron transport chain [24]. However, while the efficacy of FA is well documented, its effects on honeybee health and behavior are less studied.

The few studies that do address the effects of FA verified an increase in worker and queen bee mortality, however these results were linked to overuse of FA due to higher than recommended evaporation rates, highly resistant *Varroa*, or application to an already weak hive [25–27]. The impacts besides mortality of FA application on a healthy hive, however, remains unclear.

Foraging is a core behavior performed by *A. mellifera* in which workers gather necessary nutrients from the nectar and pollen of flowers. Foraging has a direct impact on colony productivity, health, and pollination efficacy [28]. Therefore, foraging behaviors can provide insights to potential influences of FA on a hive. Bees are very sensitive to their environment, and changes in their foraging behavior are strong indicators that the hive might be experiencing stressors, even when those are not directly causing mortality [29]. For example, pesticides or environmental contaminants have been found to cause worker bees to speed up their natural development, forage earlier in life, and have a more difficult time learning behaviors [30–32]. Worker bees that engage in precocious foraging complete far fewer foraging trips, have a higher risk of death and the hive usually experiences long-term productivity declines [33]. In general, foraging patterns "can serve as an effective and convenient tool for hive management" [28]. For all these reasons, foraging patterns can be used as a proxy to understand overall hive health.

However, quantifying bee foraging behaviors can be difficult. Often, studies measure the incoming and outgoing of bees to their hives to understand the rate of

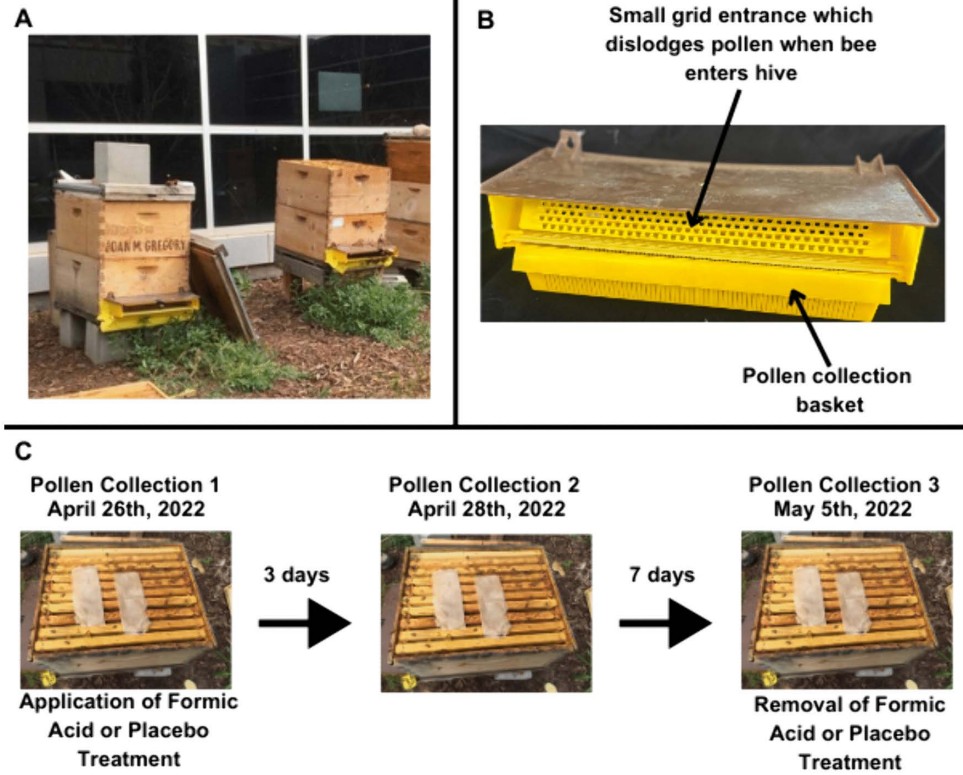

**Fig 1. Pollen collection methods and timeline. (A)** Photo of a honeybee hive with pollen trap at the entrance. **(B)** Diagram of the pollen trap. These traps are designed to remove the pollen grain from the bees' corbicula once they return from a foraging trip by forcing workers to enter the hive through a small grid that dislodges the pollen, which then falls into the collection basket. **(C)** A timeline of pollen collections during the Formic Acid or control treatment period. The photo depicts the placement of the pads in the hive. Half of the pads were soaked with Formic Acid and the other half were lightly soaked with water as a placebo. Pollen was collected three times: (1) April 26, 2022, before treatment application, (2) April 28, 2022, three days after treatment, and (3) May 5, 2022, seven days later, immediately before treatment removal.

foraging [34]. To measure the actual act of foraging (i.e., which flowers bees are visiting), researchers have traditionally relied on pollen identification or lengthy and involved fieldwork, both of which require considerable palynological expertise and are time consuming [35]. More recently, approaches utilizing information from environmental DNA are being used in place of the traditional methods [36,37]. One of these approaches, called DNA barcoding, utilizes short, specific regions of DNA that have low intraspecific variation (but high interspecific variation) to identify taxonomic presence within samples [38]. In the case of plants, DNA barcoding is performed with environmental samples that contain foraged pollen, such as pollen traps or honey.

Individual pollen samples collected from honeybees or hives usually contain DNA from more than one plant species. This multispecies sample can be assessed by expanding the DNA barcoding approach with a technique called DNA metabarcoding [39]. Pollen metabarcoding allows ecologists to quantitatively assess different species within and across samples without the need for extensive knowledge of palynology or elaborate fieldwork experiments. DNA metabarcoding of pollen has been shown to detect up to one third more species than other methods, with an estimated 70–90% plant taxa identified to species level [40,41].

In this research, we focus on DNA metabarcoding using a third-generation portable sequencer from ONT (Oxford Nanopore Technologies) (MinION Mk1C). Nanopore sequencing is a cutting-edge technology that detects differences

in electrical current as DNA is fed through a nanopore protein [42]. This technology is advantageous as it allows for an entirely in-house methodological pipeline and for long reads (kilobase-length) to be sequenced, permitting the entire trnL (UAA) intron c-d region to be obtained in a single sequencing run. To our knowledge, this research is the first to use any nanopore sequencing of the trnL (UAA) chloroplast intronic region.

In this study, we successfully provide insight into the effects of FA on *A. mellifera* foraging patterns with the use of nanopore sequencing of pollen via DNA metabarcoding to answer an ecologically relevant question.

## Materials and methods

We investigated the effect of Formic Acid (FA) treatment on *Apis mellifera* foraging behavior using six hives located at the University of Utah campus in Salt Lake City, Utah, USA. We leveraged metabarcoding technologies on the collected pollen to understand foraging patterns under different FA treatments.

### Experiment design

**Hive location and sampling.** We utilized six hives, maintained by the University of Utah Beekeepers Association, situated at two different locations: four hives near the Kahlert building (40.766374, −111.837801) and two hives near the Health Science Education Building (HSEB) (40.769540, −111.834158). Hives from both groups were selected to be either treated with FA or be a control. The Kahlert hives (denoted by first letter 'K') corresponded to two controls (Km, Kn) and two FA (Ke, Ks) hives, while HSEB hives (denoted by first letter 'H') corresponded to one control (Hm) and one FA hive (Hw). Kahlert and HSEB hives were 0.47 km apart. *A. mellifera* is an invertebrate species which does not require ethical review or approval according to the Animal Welfare Act (AWA) of the United States Code (Title 7 Agriculture; Chapter 54 Transportation, Sale, and Handling of Certain Animals; Sections 2131–2159) [43]. Permission for all sample collection was obtained from the owners, the University of Utah Beekeepers Association. Given the hives were located on a University campus, no other permits were required for sample collection.

**Pollen collection.** Pollen was collected on three occasions: (1) immediately preceding FA or placebo treatment (4/25/2022), (2) three days after applying FA or placebo treatment (4/28/2022), and (3) on the final day of FA or placebo treatment (5/5/2022) (Fig 1).

Pollen was collected with gridded traps placed at the entrance of each hive (Fig 1). Traps were designed to remove the pollen grain from the bees' corbicula once they returned from a foraging trip by forcing workers to enter the hive through a small grid, effectively dislodging the pollen, which then falls into a collection basket. To prevent workers from attempting to enter through other available openings, those were sealed with duct tape. During times when pollen was not being collected, the grid trap was raised.

Before each pollen trap collection, collection baskets were sterilized and lined with aluminum foil to minimize any cross-contamination from previous uses. At the start of pollen collection, the grid trap was lowered. After approximately 120 minutes, pollen was collected from the basket and the grid trap was once again lifted. Exact times for the collection of pollen from pollen traps in the study can be found in S1 Table. Pollen collected from the baskets were immediately transferred into sterilized plastic vials, labeled, and stored at −20°C.

**Formic acid and control treatment.** FA was applied to the hives in the form of Mite Away Quick Strips®, a FA polysaccharide gel strip. As suggested by the manufacturer, two strips were placed between the bottom and second-bottom super, the boxes that build the honeybee housing structure, to allow for maximum natural fumigation as the acid diffuses upwards through the hive (Fig 1). Typically, treatment should be removed in 7 days, but rainy conditions prohibited removal of treatment until 10 days after placement. Control hives received lightly water-soaked pads as placebos, placed in the same manner. Following treatment, all hives were monitored to confirm the queen's survival.

## DNA extraction

DNA was isolated from collected pollen samples using the Qiagen DNeasy® Plant Pro Kit. To prepare the sample for DNA extraction, 1 gram of pollen was diluted in 2 mL of lysis solution in a 50 mL tube. Exine disruption was achieved by vortexing 200 mg of pollen in a glass bead tube for 15 minutes. DNA extraction then proceeded as per the manufacturer's protocol with the slight modifications of adding 150 µL more APP Buffer than recommended and using only 50 µL of EB Buffer for final elution, the minimum amount recommended.

DNA extraction was successfully performed on a total of 29 samples, 13 samples (eight in the FA group, five in control) were taken before any treatment was applied. Six samples (four in the FA group, two in control) were taken three days after the placement of treatment. Lastly, 10 samples (six in the FA group, four in control) represent data from pollen collected at the end of treatment, 10 days after the initial application.

Final DNA concentration was assessed using a Thermo Fisher Scientific™ NanoDrop. A ratio above 1.8 DNA absorbance at 260nm/280nm is recommended by the manufacturer for sequencing, although numbers close to approaching the 1.8 threshold were used in this study since these samples experienced sequencing success. DNA concentration and exact DNA absorbance for each extraction can be found in S2 Table. Three replicates from the same pollen collection sample were completed when possible. If the collected samples did not yield enough pollen for three replicates, the maximum number of replicates possible was performed based on the available pollen. Lastly, to limit bias in subsequent steps, all DNA extractions were diluted or concentrated to 180 ng/µL.

## Molecular methods

**PCR with indexed barcodes.** DNA samples were individually amplified through PCR, targeting the *trnL* (UAA) c-d chloroplast intronic region, with forward primer 5'-CGAAATCGGTAGACGCTACG-3' and reverse primer 5'-GGGGATAGA GGGACTTGAAC-3'. Fragment length of this region is highly variable, usually ranging from 250 bp to 750 bp [44].

To allow for multiplexing, primers were modified by attaching unique 24 base pair indexes to four forward and reverse oligo primers (S3 Table). This resulted in a total of 16 uniquely indexed PCR amplicons to be pooled for subsequent sequencing and demultiplexing.

PCR reactions of extracted DNA from pollen collections (n = 29) were performed in a total volume of 31 µL containing 15 µL of Thermo Fisher Scientific™ Phusion Hot Start Master Mix, 5 µL of nuclease free water, 5 µL of DNA, 3 µL of 5 µM forward, and 3 µL 5 µM reverse primers. PCR conditions consisted of initial denaturation at 95°C for 2 minutes, followed by 20 cycles of denaturation at 95°C for 30 s, annealing 50°C for 30 s, and extension at 72°C for 1 minute, concluded by a final extension at 72°C for 2 minutes. While other studies amplifying the *trnL* (UAA) c-d chloroplast intronic region used 30–35 cycles, our preliminary testing of PCR product concentration through gel-electrophoresis found that 20 cycles produced enough amplicon product for nanopore sequencing [44].

Amplicon concentration was measured using a Qubit™ dsDNA HS Assay Kit and the Qubit™ 4.0 fluorometer and fragment length from PCR reactions were also run on a gel-electrophoresis. Amplicons were then pooled in equimolarity. In order to adequately sort samples with unique indexing combinations, we divided PCR amplicons into two pools: Pool 1 contained 16 multiplexed samples and Pool 2 contained the remaining 13 multiplexed samples. PCR clean-up was performed on both pools using a homemade mixture of AMPure XP beads. The cleaned amplicons were eluted in 30 µL of nuclease-free water and stored at −20°C.

**Library preparation and sequencing.** DNA libraries were prepared using an in-house pipeline in accordance with ONT protocols. Library preparation first consisted of end-repair and A-tailing reactions using the NEBNext® Companion module for ONT Ligation Sequencing. Adapter ligation and loading preparation were performed using ONT Ligation Sequencing Kit V13, SQK-LSK110. Pool 1 was processed to be sequenced on a MinION Flow Cell (R9.4). Pool 2 was processed to be sequenced on the MinION Flongle Flow Cell (R9.4). Sequencing was performed on the ONT MinION Mk1C device. Experiments were run for 48 hours or until pore exhaustion.

## Bioinformatics

**Basecalling.** Initial basecalling occurred using ONT MinKNOW (v20.06.2), producing real-time results for initial confirmation of sequencing read lengths of *trnL* barcode and other initial quality metrics. We then re-basecalled the sequences using ONT's higher fidelity basecaller (SUP) with Dorado (v0.8.0) to limit error rates in the reads and maximize read output (S4 Table). After re-basecalling, reads were categorized as pass or fail based on a Phred quality score threshold of eight. NanoPlot (v1.19.0) [45] was run on the re-basecalled sequences to confirm read quality and read length distribution.

**Demultiplexing and denoising.** Reads were demultiplexed into their respective original samples using minibar (v0.24) [46]. To optimize the number of reads successfully demultiplexed into samples, various editing distances for the indexes and primers were evaluated. A balance between maximizing the number of sorted reads while minimizing unknown or multiple matched reads occurred at an allowed editing distance of 9 ("-e 9") (S5 Table). Indexes and primers were removed once demultiplexing was complete.

Next, long and short reads were filtered out to reduce the presence of errors from sequencing, remove concatenated reads, and ensure sequences accurately represented the *trnL* c-d region. To account for significant length variability of the target region [44], we excluded reads longer than 1050 bp and shorter than 150 bp, using *USearch* (v 2.15.0) [47] (S6 Table).

**Database curation.** To create a *trnL*-specific database, we used the protocol constructed by Robert Greenhalgh [48]. Using the search terms "biomol_genomic[PROP] AND is_nuccore[filter] AND chloroplast[filter]" within the entire NCBI Blast database, a collection of chloroplast-only sequences was compiled. This chloroplast database was further refined to include only *trnL* sequences by filtering to include only sequences containing the forward 5'-GGGCAATCCTGAGCCAA-3' and reverse 5'-CCATTGAGTCTCTGCACCTATC-3' primers. The final database contained 18,462 sequences, each representing a unique taxonomic unit.

**BLAST.** We used NCBI's blastn+ (v2.15.0) [49] to verify sequence identity for all obtained sequences. Considering effects of plant morphospecies on barcoding identity, taxonomic identity was assigned to genus level so then all reads could be incorporated in our analyses. All top matches (best bit score) were included regardless of percent identity match or e-value [50]. To limit the impacts of shallow read-depth, reads that made up less than 1% of a sample's total reads were removed [51,52].

## Statistical methods

Before performing statistical analysis, normalized Relative Read Abundance (RRA) was calculated for each plant genus per sample. RRA data was used in all following statistical analyses. All analyses and figures were generated using different packages (referenced below) in R version 4.4.1 [53] in Rstudio [54]. Data manipulation was done using the dplyr package [55]. P-values of $< 0.05$ were considered statistically significant. Pollen collected from hives in the same location, receiving the same FA or placebo application were treated as biological replicates. Additionally, three distinct subsamples from each pollen collection were sequenced when possible.

Relative read abundances by treatment were visualized with a stacked bar chart and heatmap from ggplot2 (v3.5.1) (Figs 2 and 3) [56]. To identify differences in the data between treatments and location, we used multivariate analyses. Results were separated in the respective six treatments: Pre-Formic Acid, Pre-Control, During-Formic Acid, During-Control, Post-Formic Acid, and Post-Control. First, we transformed the data using Hellinger-transformation (herein "hellinger_transformed_data") to reduce the influence of high or low abundance plant genera. All following calculations were performed with this transformed data (herein "data"). We then calculated distance matrices between plant genera using the "vegdist" command from the vegan package (v2.6.4) [57]. To understand the dispersion effect on the distance data (herein "distance_data") we used the "betadisper" command from vegan, with "type = centroid". A permutational analysis of variance (PERMANOVA) was performed on the distance data to quantify the relationship between the different treatment groups and/or location ("treatment" / "location"). The following structure was used for this calculation:

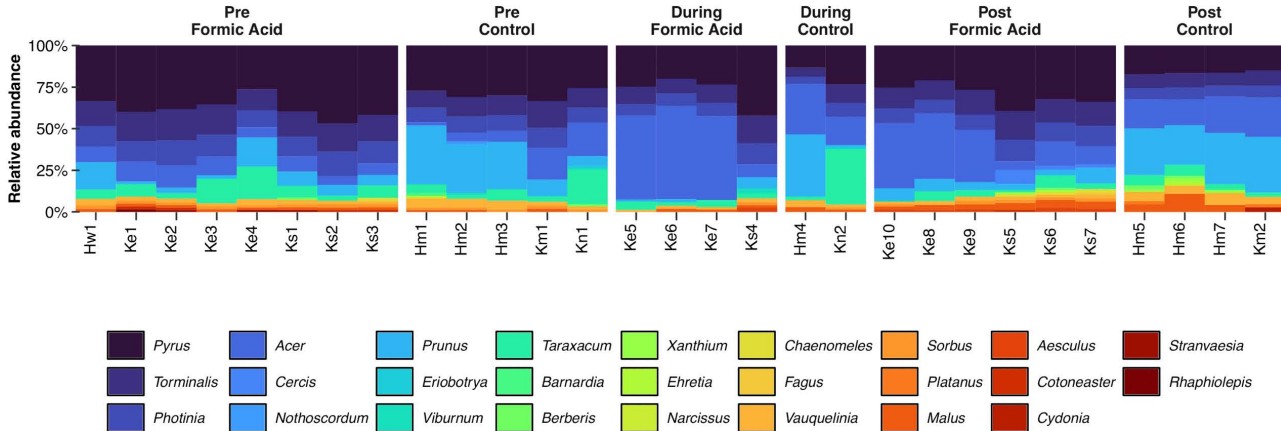

**Fig 2. Relative read abundances of taxonomic composition from individual pollen samples throughout treatment application.** Relative abundance bar chart showing the taxonomic composition of plant genera identified from DNA sequences of pollen samples. The x-axis represents individual Sample IDs of different pollen collections: different locations of the hives on the University of Utah campus are marked with either an 'H' or an 'K', and the following number denotes the sample number (e.g., Ke1, Ke2, and Ke3 are replicates from the same hive). Each color represents a different plant genera identified in the pollen, and the size of the color for each sample indicates the relative abundance of the plant genus in relation to the entire sample. The chart is organized by different treatment groups, hives that received Formic Acid treatment and hives that received the control (placebo) treatment and is organized in the order the pollen was collected to allow comparison over time.

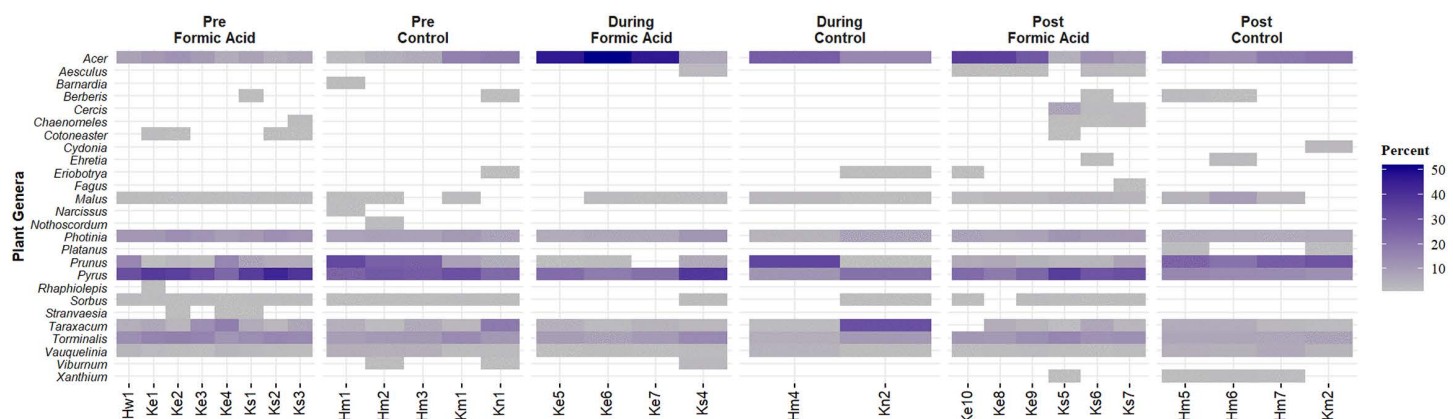

**Fig 3. Heatmap of plant genera detected from individual pollen samples throughout treatment application.** Heatmap illustrates the change in presence of plant genera per pollen sample across different collection dates. Color gradient changes from blue (for a higher percentage presence of a plant genus in a sample) to grey (a lower percentage presence). The chart is organized by different treatment groups (Formic Acid vs control) over time ('Pre', 'During', and 'Post').

**adonis2(distance_data~as.factor(data$treatment), data = distance_data, permutations = 9999)**

Since the result was significant, pairwise post-hoc testing was then performed using the RVAideMemoire package (v0.9.83.1) [58] to identify potential significance between specific treatment groups (Pre-Formic Acid, During-Formic Acid, Post-Formic Acid, Pre-Control, During-Control, or Post-Control) or hive location (Hm, Hw, Ke, Km, Kn, or Ks). The following structure was used for these calculations:

**pairwise.perm.manova(distance_data, hellinger_transformed_data$treatment, p.method = "holm")**

To visualize orientational differences between treatment groups, we applied the unconstrained, Non-Metric MultiDimensional Scaling (NMDS) modeling from the vegan package to the Hellinger-transformed data using the "metaMDS" command from vegan (v2.6.4) [57] (Fig 4).

Bar charts were created to further visualize and understand differences between groups (Fig 5). Diversity metrics (including Shannon Diversity Index (SDI) and plant genera richness) were calculated using the R vegan package (Fig 6). Shapiro-Wilks tests were carried out to verify normality. Next, a one-way ANOVA was performed, followed by a post-hoc Tukey HSD testing when significance was detected. All graphs were created using the RStudio package ggplot2 (v3.5.1) [56].

## Results

### Pollen collection and DNA extraction

Pollen was successfully collected from all six hives. Throughout the treatment process, no hive lost its queen. DNA extraction and PCR were successfully performed on a total of 29 pollen samples.

### Sequencing and taxonomic identification

In total, ONT nanopore sequencing generated 10,300,000 reads using the MinION Flow Cell and 277,900 reads with the MinION Flongle Flow Cell. After performing quality control, which involved removing reads with a Phred Q-score below eight (8) and conducting SUP basecalling, 8,317,865 (80.8%) and 221,101 (79.6%) reads remained, respectively.

These reads were successfully sorted into the 29 pollen samples. After analysis of editing distances, an editing distance of 9 sorted the maximum amount of reads into their corresponding sample (S5 Table). High-fidelity basecalling also strongly impacted the number of reads sorted (S4 Table).

After demultiplexing and denoising, each of the 29 samples contained a mean of 109,487 reads, a median of 91,602 reads, summing to 3,175,147 total demultiplexed reads, corresponding to 38% of the initial passed reads.

In total, 26 distinct plant genera (Fig 2) were identified from 10 different families (S7 Table). Honeybees most often visited five genera, based on pollen abundance: *Pyrus* (Rosaceae), *Torminalis* (Rosaceae), *Photinia* (Rosaceae), *Acer* (Sapindaceae), and *Prunus* (Rosaceae).

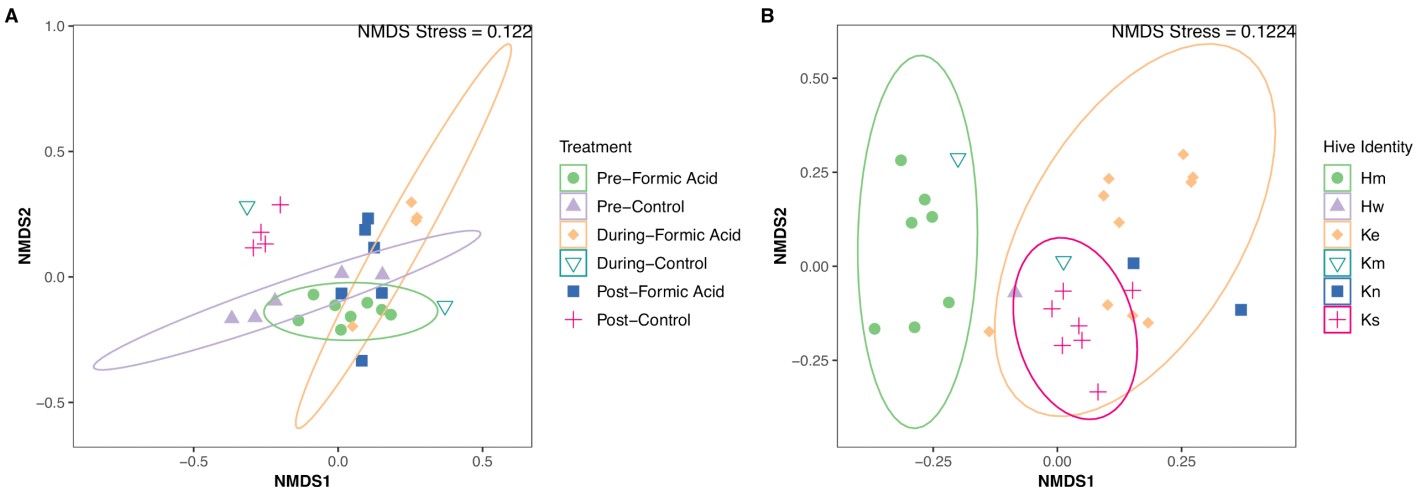

**Fig 4. Non-metric multidimensional scaling (NMDS) ordination plots illustrate differences in foraging choices based on (A) treatment groups and (B) individual hives.** Distances were calculated using the Hellinger dissimilarity index. In these plots, points that are closer together represent more similar pollen compositions. Ellipses enclose a 95% confidence limit and NMDS stress values indicate how well the visual representation preserves the original distance relationships. Stress values below 0.2 indicate a good fit, meaning the NMDS plot reasonably represents the original data structure.

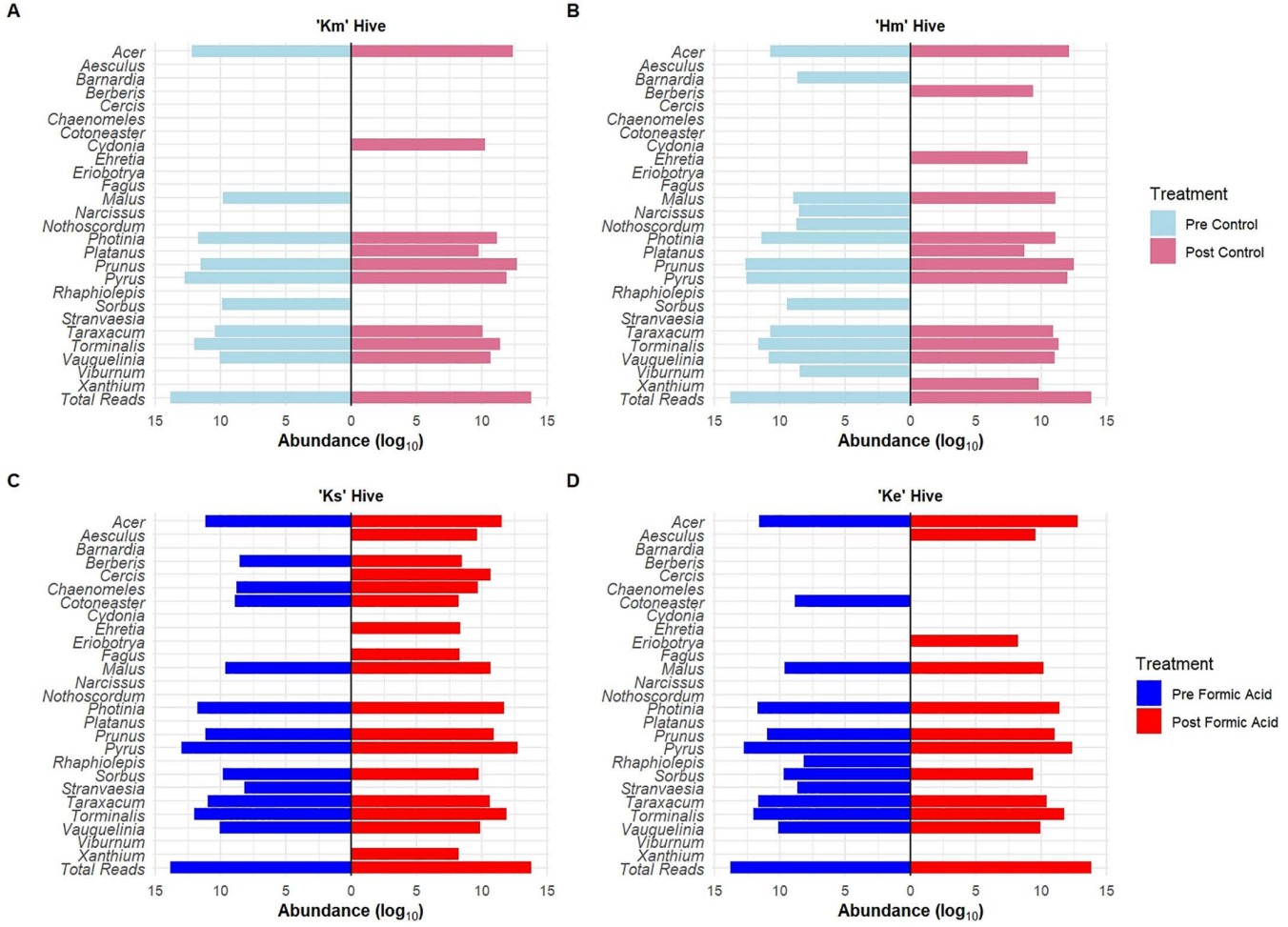

**Fig 5. Bar charts showing the taxonomic presence of plant genera identified from DNA sequences of pollen samples collected from honey-bees after foraging.** Each of the plots represent individual honeybee hives before and after Formic Acid or control treatment. Hives **(A)** 'Km' and **(B)** 'Hm' represent the control group, while hives **(C)** 'Ke' and **(D)** 'Ks' receive Formic Acid treatment. For each hive, the left side of the plot represents pollen collected before treatment, and the right side represents pollen collected at the end of the treatment period. Bar length corresponds to the relative read abundance of each plant genus, $log_{10}$-normalized. The y-axis lists all plant genera identified in the study. Difference between the left and right sides of each plot indicates changes in genera identified from the pollen following treatment.

The average percentage identity match from blastn+ was 93.6%, the highest match was 100% and the lowest match was 72% (which only occurred twice throughout the entirety of the dataset). By removing the bottom 1% of identified genera in each sample, 406,100 (12.8%) of reads were excluded.

On average, each sample contained 10 plant genera. The highest number of plant genera detected within a single sample was 14 from the Ks6 hive during the Post-Formic Acid treatment. The lowest number of plant genera detected within a pollen sample was seven by the Ke5 and Ke7 hives during the Formic Acid treatment.

## NMDS plots

Treatments (Pre-Formic Acid, During-Formic Acid, Post-Formic Acid, Pre-Control, During-Control, or Post-Control) did not differ significantly in dispersion (PERMDISP, F = 2.5581, p < 0.001). We found a significant difference between the treatments (PERMANOVA, F = 3.9824, p = 0.001, $R^2$ = 0.46402) (Fig 4A).

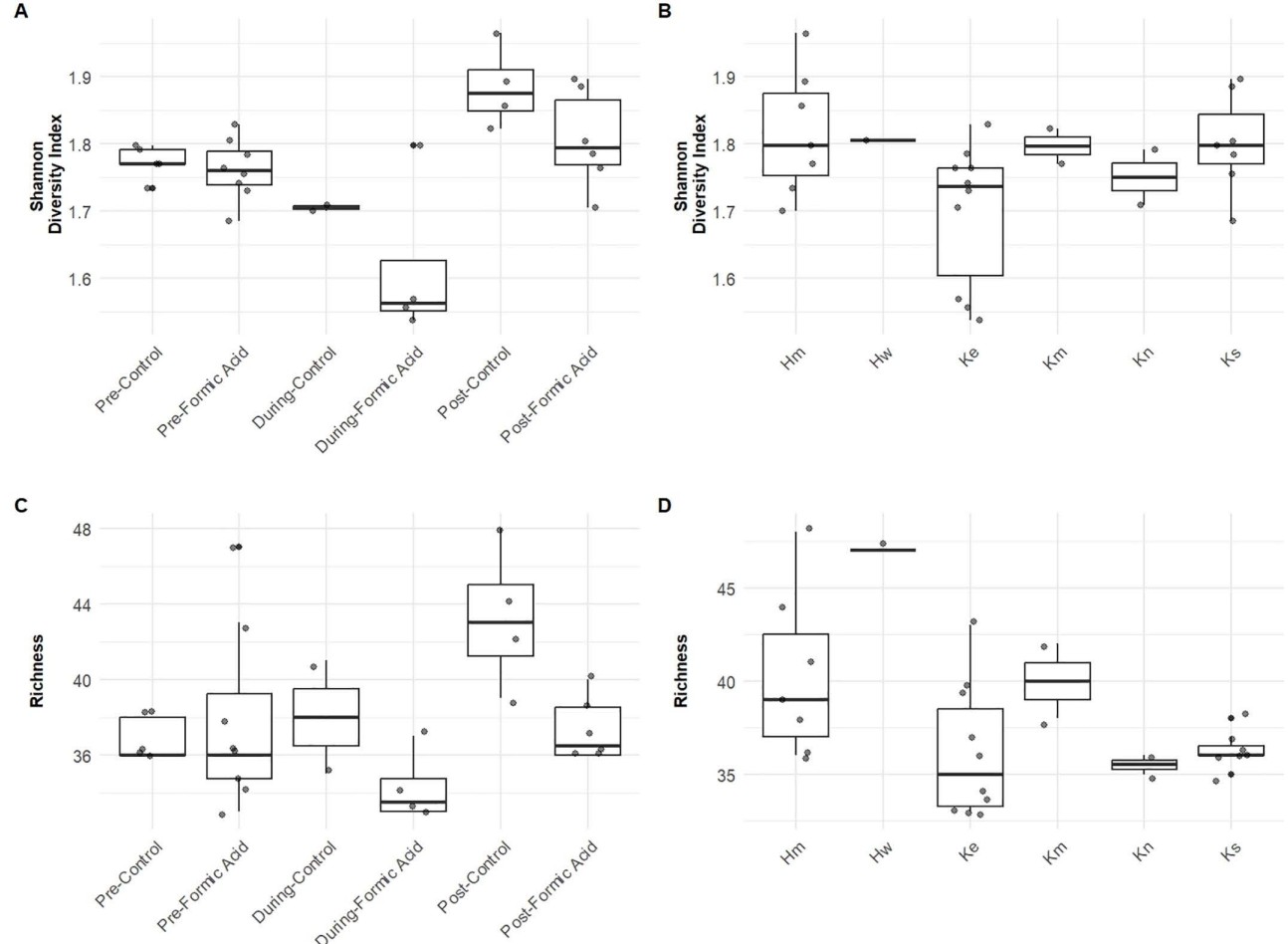

**Fig 6. Shannon diversity index (SDI) and plant genera richness by treatment or individual hives.** Box plot diversity measurements are calculated using plant genera data identified from DNA sequences of pollen collected from honeybees under different treatments and from individual hives ('Hive ID'). The dark line at the center indicates median value, the top and bottom of the box indicate 1st and 3rd quartile, respectively. Each dot is the calculated value of an individual pollen collection sample. The top row depicts the SDI (higher the value, lower overall diversity) between plot (A) the two treatment groups, Formic Acid Treatment and control hives before, during and post treatment and plot (B) by Hive ID in which all samples collected from each hive are included. The bottom row represents richness diversity, (the number of plant genera present in the group) of the two treatment groups, plot **(C)**, and richness by Hive ID, plot **(D)**. See S9 Table for comprehensive statistics extracted from boxplots in this figure. See S12 Table for post-hoc analysis results..

Hive location (Hm, Hw, Ke, Km, Kn, or Ks) differed significantly in dispersion (PERMDISP, F=7.3106, p<0.001, location=0.0003142). We also found significant differences within hive location (PERMANOVA, F=4.8198, p=0.001, $R^2$=0.51167), although this outcome is partially influenced by the significant difference in dispersion across hive locations (Fig 4B).

Pairwise comparisons of both treatments and location did not reveal any significant differences (S8 Table).

### Type and size of differences: bar charts and diversity indexes

To further compare treatment groups, back-to-back bar charts were constructed using the four hives with the highest number of replicate data in both the pre- and post-treatment groups were selected for comparison ('Km', 'Hm', 'Ks', and 'Ke') (Fig 5).

There were notable differences between plant genera before and after treatment application in both FA hives and control hives (Fig 5A). For example, in hive 'Hm' (control), four genera (*Barnardia, Narcissus, Nothoscordum, Sorbus*) were detected pre-treatment but not post-treatment. Similarly, for the same hive, four genera (*Berberis, Cydonia, Photinia, Xanthium*) were identified in the post-treatment but absent pre-treatment. The other control hive, 'Km' exhibits similar pre- and post- groups differences (Fig 5B).

The hives treated with FA ('Ks' and 'Ke') showed a comparatively higher number of plant genera differences between the start and end of treatment (Fig 5 C-D). For example, in hive 'Ks', five plant genera (*Aesculus, Cercis, Cydonia, Fagus, Xanthium*) were detected post-treatment but absent pre-treatment. For hive 'Ke', two genera (*Aesculus, Eriobotrya)* were detected post-treatment by absent pre-treatment.

The Shannon Diversity Index (SDI) and overall richness suggests changes in plant genera diversity visited throughout the sample period (Fig 6). SDI values revealed a shift from higher diversity to lower diversity over time, spanning the pre- (median of 1.75–1.77) during- (1.56–1.70), and post- (1.79–1.87) treatment periods. Notably, these shifts in median diversity occurred similarly in both control and FA treatment groups, although the control group shows a higher SDI diversity by the Post-treatment period. In terms of overall plant genera richness, post-control hives had comparatively higher richness than any other sample. Performing post-hoc Tukey HSD on SDI results sorted by treatment groups revealed five significantly different pairs, which showcased differences between the Pre, During, and Post samples (S12 Table). When the same post-hoc tests were performed on richness results by treatment groups, only Post-Control and During-Formic Acid differed significantly from one another (adjusted p-value = 0.01).

No clear, identifiable changes in plant diversity were observed for hives in different locations (SDI median ranged from 1.73–1.80 for all locations) (Fig 6). When comparing hives from the HSEB location (hives beginning with 'H') to those from the Kahlert location (hives beginning with 'K'), there was no clear trend or significant differences in central tendency, as indicated by overlapping quartile boxes. SDI medians for hives Ke and Kn differed only marginally, with 1.74 and 1.75, respectively. These values slightly contrast with hives Km and Ks at the same location (both had medians of 1.80). The results of a one-way ANOVA test on SDI results sorted by hive location were not significant (adjusted p-value = 0.138). When assessing richness results sorted by hive location, a significant ANOVA and the following post-hoc Tukey HSD testing revealed a significant relationship between hives Ke and Hw throughout the entire pollen collection period (adjusted p-value = 0.042).

## Discussion

Honeybees (*Apis mellifera*) are very sensitive to their environment and changes in foraging behavior can indicate biotic and abiotic challenges [28]. By analyzing differences in their foraging patterns over time, we can provide valuable insight into the impacts of different inputs into honeybee environments [29]. In this study, we used DNA metabarcoding to assess how the common *Varroa destructor* (hereafter *Varroa*) mite treatment of Formic Acid (FA) influences *A. mellifera* foraging behavior (n = 6 hives). We found that there are foraging differences between hives that received FA treatment and the hives that received a placebo treatment (Fig 1 and 4). We also found that individual hives had unique forage preferences (Fig 4B). Under our sampling design, our findings suggest that FA treatment is associated with changes in foraging behavior but not sufficient to overcome inherent hive-specific foraging preferences and therefore FA is unlikely to be meaningfully detrimental. The detection of distinct hive foraging "identities" validates pollen metabarcoding as a foraging behavioral tracking tool, demonstrating a remarkably accessible and novel approach using Oxford Nanopore Technologies (ONT) platforms.

### Impact of formic acid on hive foraging

Hive infection from the external parasite *Varroa* is detrimental to the health of a hive [13,17]. When applied properly, FA is an effective tool used by many beekeepers for mite control, but it can have impacts on hive health [59]. In our study

we detected differences between the FA and control hives across multiple analyses. Specifically, our non-metric multidimensional scaling (NMDS) analysis highlighted a difference in foraging before and after treatment, a finding reinforced by our significant PERMANOVA result (p = 0.001) (Fig 4A and S8 Table). It is important to note that data homogeneity, as assessed through dispersion measurements (PERMDISP), confirms that these patterns are not due to data asymmetry but to actual differences in foraging.

Initial insights into the magnitude of the effect of FA on foraging preferences can be drawn from our findings. For example, the lack of significance in the post-hoc pairwise tests, despite a significant PERMANOVA, indicates a more global variance between the data centroids rather than specifically between individual treatment groups. Additionally, post-hoc Tukey HSD tests of Shannon Diversity Index (SDI) values revealed significant differences; however, these differences occurred between pollen collections (pre-, during-, and post- FA or placebo treatment) rather than between treatment groups (S12 Table). Together, these results suggest that any hive disturbances, FA or placebo, may alter foraging behavior, although such changes may occur gradually. Further research incorporating a longer sampling period may help clarify whether our detected patterns reflect gradual responses to FA exposure or differences in resilience following disturbance.

There is well-established evidence that honeybees are impacted by toxic contaminants introduced into the hive [30–32]. Detected differences following FA application supports previous research suggesting that substances introduced into *A. mellifera* hives likely affects the hive's behavior [26]. As beekeepers continue to combat *Varroa* infections, it is crucial to recognize that even effective treatments are sensed by the hive and might have the potential to trigger unexpected, detrimental outcomes. Analyzing the factors that promote colony resilience and adaptation to disturbances could help determine when a hive is most amenable to the introduction of external substances. Additional studies would help to refine FA applications to maximize *Varroa* mortality while minimizing its impact on hive health and homeostasis. Our research poses a predicament for beekeepers: for now, it might be advised to use alternating treatment methods (e.g., switching between oxalic and FA), limiting the dosage of treatment, or only treating a hive when *Varroa* infection has been detected instead of as a preventative measure [59].

When assessing the size or cause of the observed changes in foraging over time, it must be considered within the greater framework that honeybees are complex and their hives can act as a superorganism dependent on many factors, not just nutritional needs [60]. While we observed shifts in plant genera richness visited by the FA and control hives, diversity of foraged pollen alone does not necessarily indicate poor health of the hives as it could also indicate a change in nutritional needs due to other reasons or plant phenological changes [61]. Ultimately, our results are a reminder that detecting impacts of FA might require that other factors influencing *A. mellifera* foraging choices be considered.

## Hive foraging identity

Our results support the idea that individual hives have distinct foraging preferences. *Apis mellifera* hives are known to develop individual foraging preferences due to many factors such as colony size, strength, and brood rearing activity [34,62]. If biotic conditions are maintained, specifically in regard to the access to floral resources, it would be assumed that hives forage identically. Interestingly, our NMDS plots (Fig 4B) revealed a tendency for hives to have distinct preferences, even when located in close proximity. Our significant PERMDISP test did indicate that our hive location data differed in variability, meaning the significant PERMANOVA result (p = 0.001) is ambiguous. Subsequent PERMANOVA post-hoc pairwise comparisons between hive locations did not reveal significant differences (S9 Table). This suggests that the observed hive identities in our NMDS plots should be interpreted cautiously, as an initial indicator to spark further inquiry. Consequent research with a larger sample size and a longer sample collection period would hopefully tease apart these differences.

Previous studies have looked at how distinct hives forage in response to changes in nutrition or individual behaviors, but there is a paucity of information describing the innate presence of a hive foraging identity, especially over long time periods (such as the lifetime of a queen) [63–66]. Various factors could influence this identity such as genetics,

hive-specific environmental stressors, and queen-related factors [34,62]. Further research into understanding the concept of hive foraging identity, particularly in relation to floral availability and long-term foraging choices, could provide valuable insight into the complexities of *A. mellifera* decision-making. This could help answer questions linked to how, and to what degree, foraging is driven by access to floral resources, nutritional needs, responses to external impacts, or familiarity with known visited resources.

To our knowledge, this is the first time that individual hive foraging preferences have been examined through DNA metabarcoding. Our detection of individual hive foraging identity highlights the discriminatory power of pollen metabarcoding and its potential as an emerging methodology for detecting ecological patterns.

## Pollen metabarcoding

Our study utilized an emerging novel approach to pollen metagenomics and nanopore sequencing. While an increasingly robust body of literature has established the validity of DNA metabarcoding methods, they are only recently beginning to be applied to ecological research [36]. In general, there is a lack of consensus on the ideal DNA barcoding region from plants. For plant barcoding, the region's universality, strong sequence-to-pollen prediction, and discriminatory power makes *trnL* (254–767 bp) a good candidate [44,65–67].

Until recently, research involving DNA sequencing has relied on access to specialized labs with technical and expensive machinery that require powerful computers. Nanopore sequencing could be a tool to decrease these barriers and facilitate the adoption of metabarcoding approaches by providing unparalleled accessibility and real-time sequencing (or to paraphrase ONT's motto: "analysis of anything, by anyone, anywhere") [68]. High-throughput methods, such as ONT, have begun to eliminate this barrier, opening the doors of DNA-based approaches for communities and individuals that otherwise wouldn't be able to do this research [69]. Our research is a prime example of the accessibility of nanopore sequencing. All bioinformatic analyses were conducted by the first author (CLW) while an undergraduate student at the University of Utah. Additionally, since an entire experimental protocol using ONT can be conducted in-house, it can be much more conducive to the creation of personalized pipelines following individual needs (e.g., type of basecalling algorithm, output format, local troubleshooting). As noted by Bell [36], bioinformatic steps can often be the highest barrier to completion of DNA sequencing projects. During the completion of our study, CLW heavily relied on many available resources as part of ONT's Nanopore Community and benefited from the in-house pipeline in her troubleshooting.

Nanopore sequencing allowed us to sequence the entirety of the *trnL* region, a DNA fragment too long for some alternative sequencing methods, like Illumina (~600-bp paired-end reads on the Illumina MiSeq platform). Combining more than one plant barcoding region in future pollen DNA metabarcoding studies, such as rbcL, matK, ITS1, or ITS2, could enhance species-level identification, further refining our understanding of honeybee foraging behavior [70,71].

Studying changes in pollen collection by bees has been traditionally done through palynology, which can be incredibly time consuming and require a high level of taxonomic expertise [72]. While it has been proposed these barriers could be overcome with artificial intelligence-based analysis, some pollen lack morphological differences, thus inherently limiting visual taxonomic resolution [73,74]. DNA metabarcoding methodologies routinely detect higher species counts than visual observations [75].

Still, DNA sequencing currently has many limitations [36], so we don't advocate for the replacement of visual pollen identification. For example, it is still not fully understood how DNA processing techniques such as extraction and PCR can cause a disconnection between sequence reads and actual taxonomic abundance in a sample [65,76]. The complementarity of both methods might provide increasingly robust knowledge, and help refine metabarcoding accuracy. Another limitation to sequencing is the lack of comprehensive databases for the various different plant barcoding regions. We hope that further use of ONT sequencing and the *trnL* gene will contribute to extended databases that could improve taxonomic discrimination when focusing on that particular barcoding region.

In conclusion, our findings indicate that FA treatment is associated with differences in foraging behaviors of *A. mellifera* but, due to the presence of hive identity and the inherent complexity of understanding decision making by *A. mellifera*, the impact does not warrant discontinuing its use. Given the widespread use of FA treatment, particularly in commercial settings where it effectively targets *Varroa* mites, our results are reassuring. We also found that individual hives have specific foraging preferences, supporting the discriminatory power of DNA metabarcoding using ONT and the *trnL* barcoding region. Delineating the specific drivers of *A. mellifera* foraging choices would provide critical insights into how human-mediated variables alter hive dynamics and resilience. The continued use of nanopore sequencing methods to ecologically relevant questions is an exciting avenue and has the potential to open many opportunities for scientists at different career levels, technical capabilities, and degrees of funding to conduct meaningful research.

## Supporting information

**S1 Table. Times and durations of pollen collections.** Pollen was collected before Formic Acid (FA) or placebo treatment application (4/25/2022), during treatment application (4/28/2022), and at the end of treatment application (5/5/2022). Pollen was collected using pollen traps as shown in Fig 1.
(DOCX)

**S2 Table. Pollen DNA concentrations.** Concentrations were measured with the Nanodrop instrument (Thermo Fisher Scientific™) after extraction.
(DOCX)

**S3 Table. Forward and reverse indexes.** Unique forward and reverse indexes attached to the *trnL* (UAA) intron c-d primers for the present study. The shortest hamming distance between any of the indexes is 13 base pairs and the longest is 19 base pairs.
(DOCX)

**S4 Table. Comparison of ONT's basecalling algorithms.** Number of reads retrieved from fast basecalling and high fidelity (SUP) basecalling for the MinION Flow Cell and Flongle runs conducted in this study. For sample information go to page 7 and 8 in the main body of the present study.
(DOCX)

**S5 Table. Demultiplexed read counts under different allowed editing distances.** Reads were demultiplexed (using minibar.py) with different maximum permitted editing distances of each index (see S3 Table), which is the individual identifying index placed at the beginning and end of each read. Read counts are from both minION flow cell and Flongle datasets, the most sorted reads occur at an editing distance of 9 ("-e 9").
(DOCX)

**S6 Table. DNA sequence read counts per sample before and after denoising.** Denoising consisted of removing reads shorter than 150 bp and longer than 1050 bp. Table also includes the total reads removed per sample and percentage of reads removed from the original read counts.
(DOCX)

**S7 Table. Plant genera identified from DNA sequencing (after denoising) and the corresponding family name.**
(DOCX)

**S8 Table. Pairwise PERMANOVA comparing relative read abundance data between treatments.** Significant results are highlighted in bold. Analyses were calculated using the "pairwise.perm.manova" function from the RVAideMemoire package in R and performed on Hellinger-transformed, relative read abundance data.
(DOCX)

**S9 Table. Pairwise PERMANOVA comparing relative read abundance data between hive locations.** Significant results are highlighted in bold. Analyses were calculated using the "pairwise.perm.manova" function from the RVAideMemoire package in R and performed on Hellinger-transformed, relative read abundance data. For sample information see page 7 in the main body of the present study.
(DOCX)

**S10 Table. Shannon Diversity Index summary statistics.** All relevant statistics calculated from DNA reads of plant genera collected by honeybees under different treatment conditions and locations.
(DOCX)

**S11 Table. Richness summary statistics.** All relevant statistics calculated from DNA reads of plant genera collected by honeybees under different treatment conditions and locations.
(DOCX)

**S12 Table. Significant results from Tukey HSD post-hoc tests of Shannon Diversity Index (SDI) values by treatment group.** Additional Tukey HSD post-hoc tests of SDI values by hive location and genera richness by treatment group and hive location yielded one or no significant results. These results are presented in the Results section.
(DOCX)

## Author contributions

**Conceptualization:** Claudia L. Wiese, Heather M. Briggs, Joshua G. Steffen.

**Data curation:** Claudia L. Wiese, Rodolfo S. Probst.

**Formal analysis:** Claudia L. Wiese.

**Funding acquisition:** Joshua G. Steffen.

**Investigation:** Claudia L. Wiese.

**Methodology:** Claudia L. Wiese, Rodolfo S. Probst, Heather M. Briggs, Joshua G. Steffen.

**Resources:** Rodolfo S. Probst, Joshua G. Steffen.

**Software:** Claudia L. Wiese, Rodolfo S. Probst.

**Supervision:** Rodolfo S. Probst, Heather M. Briggs, Joshua G. Steffen.

**Visualization:** Claudia L. Wiese.

**Writing – original draft:** Claudia L. Wiese.

**Writing – review & editing:** Claudia L. Wiese, Rodolfo S. Probst, Joshua G. Steffen.

## Acknowledgments

We thank all the undergraduate researchers who assisted with field data collection: **Collin Cadens, Mia Barth, Allie Perkins, and Benning Lozada**. Special thanks to **Eleanor Wachtel** for assisting with DNA extractions and **Sydney Larson** for support with coding and bioinformatics processing.

We acknowledge the **Center for High Performance Computing (CHPC)** for providing access to high-computing resources necessary for processing large datasets, as well as for their assistance.

CLW thanks the **University of Utah Beekeeping Society** for granting her access to campus hives during the experiment.

RSP thanks to **ONT Education Beta Program** for technology and reagent support.

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
