## [Decision Letter · Decision Letter 0]

4 Nov 2025

*Apis mellifera*

Dear Dr. Wiese,

Thank you for submitting your manuscript to PLOS ONE. After careful consideration, we feel that it has merit but does not fully meet PLOS ONE’s publication criteria as it currently stands. Therefore, we invite you to submit a revised version of the manuscript that addresses the points raised during the review process.

We look forward to receiving your revised manuscript.

Kind regards,

Kai Wang

Academic Editor

PLOS ONE

Journal Requirements:

4. Please amend your authorship list in your manuscript file to include author Claudia Wiese.

5. Please amend the manuscript submission data (via Edit Submission) to include author Claudia L. Wiese.

6. note that Figure 1 in your submission contain copyrighted images. All PLOS content is published under the Creative Commons Attribution License (CC BY 4.0), which means that the manuscript, images, and Supporting Information files will be freely available online, and any third party is permitted to access, download, copy, distribute, and use these materials in any way, even commercially, with proper attribution. For more information, see our copyright guidelines: http://journals.plos.org/plosone/s/licenses-and-copyright.

Reviewers' comments:

Reviewer's Responses to Questions

**Comments to the Author**

1. Is the manuscript technically sound, and do the data support the conclusions?

Reviewer #1: Yes

Reviewer #2: Yes

2. Has the statistical analysis been performed appropriately and rigorously?

Reviewer #1: Yes

Reviewer #2: Yes

3. Have the authors made all data underlying the findings in their manuscript fully available?

Reviewer #1: Yes

Reviewer #2: Yes

4. Is the manuscript presented in an intelligible fashion and written in standard English?

Reviewer #1: Yes

Reviewer #2: Yes

Reviewer #1: The manuscript titled “Effect of formic acid treatment on Apis mellifera foraging behavior using nanopore metabarcoding technologies” presents an innovative approach combining ecological observation and molecular techniques to explore the effects of formic acid (FA) treatment on honeybee foraging behavior. The integration of pollen DNA metabarcoding with nanopore sequencing (ONT MinION) represents a valuable contribution to applied apiculture and pollination ecology.

However;

The sample size (n=6 hives) limits statistical inference. This should be clearly stated as a limitation in both Abstract and Discussion.

The PERMANOVA detected significant differences between treatments, but post-hoc comparisons were nonsignificant, suggesting a broad rather than treatment-specific effect. This nuance should be discussed more explicitly.

Clarify whether biological replicates (hives) or technical replicates (DNA extractions) formed the basis of statistical analysis.

Adding effect size estimates (R² or beta-dispersion) and clarifying how variability was partitioned would strengthen conclusions.

Despite these limitations, the experimental approach is solid, and results are consistent with the hypothesis that FA treatment may modestly alter foraging composition.

Dispersion differences (PERMDISP) were significant; this assumption violation should be explicitly discussed since it affects PERMANOVA interpretation.

Diversity indices (Shannon, richness) are presented visually but could be supported by statistical comparison (e.g., ANOVA or Wilcoxon tests).

Include the R² values for treatment effects in Results and the Supplementary Tables for transparency.

Overall, statistical analysis is rigorous, though additional quantitative interpretation would strengthen the conclusions.

The figures are scientifically appropriate and clearly convey the findings; however, several appear to be low-resolution raster images embedded in the Word manuscript. To meet publication quality, all figures should be exported in ≥300 dpi or, preferably, vector-based formats (PDF, EPS, or TIFF).

Specific observations:

Figure 1 (photo + schematic): the photograph is slightly blurry, and fonts in the diagram are too small; scale or legend text should be enlarged.

Figures 2–3 (bar chart and heatmap): color palette uses similar hues, reducing contrast. Please use a colorblind-friendly palette and increase label font size.

Figure 4 (NMDS plots): acceptable but should be exported directly from R in vector format for clarity.

Figures 5–6 (diversity plots): scientifically correct but axis labels are too small; increase font and ensure legends remain legible at journal layout size.

Highlight that hive-level effects may reflect genetic, environmental, or queen-related factors.

Acknowledge that “hive identity” could also reflect temporal floral availability rather than intrinsic hive traits.

Temper statements implying causation: instead of “FA treatment has an impact,” use “FA treatment was associated with detectable shifts in foraging composition.”

Reviewer #2: The introduction has a solid theoretical fundamention, the method is replicable and well detailed, the material is well discussed, and the conclusion aligns with the objectives. The references are also relevant to the research.

**Do you want your identity to be public for this peer review?** For information about this choice, including consent withdrawal, please see our Privacy Policy

Reviewer #1: **Yes:** KEMAL KARABAĞ

Reviewer #2: No

You may also use PLOS’s free figure tool, NAAS, to help you prepare publication quality figures: https://journals.plos.org/plosone/s/figures#loc-tools-for-figure-preparation

---

## [Author Response · Author response to Decision Letter 1]

1 Feb 2026

Dear Editors,

We wish to express our sincere gratitude to the editors and reviewers that contributed to this review; we acknowledge the time and effort it took to review our paper and provide helpful and constructive feedback. We have made relevant changes to our manuscript and we hope that these revisions address the concerns of the reviewers and increase the overall quality of our submission. We want to acknowledge that protocols used in this paper are already publicly available.

In addition to submitting a revised version to PLoS ONE, in the effort for good faith and commitment to the most accurately available research, we will also submit the updated manuscript version to the pre-print online on BioRxiv.

Below we have added responses to each editor and reviewers’ comments. We have specified the line numbers where the changes were made. We hope that this updated version adequately addresses all concerns.

Sincerely,

Claudia L. Wiese

The Academic Editor wrote (Comment #1): “Please ensure that your manuscript meets PLOS ONE's style requirements, including those for file naming.”

Response to Comment #1: We have reviewed the PLOS ONE style requirements and have made the following changes:

Revised the title page by removing the article’s short title and the author’s ORCID ID’s (see Title Page).

Adjusted the level 2 and level 3 headings font size, corrected type case, and removed the indented bullets (see headings in Materials and methods, Results, and Discussion sections).

Adjusted each supplemental table name to be formatted correctly. For example writing “S1 Table” instead of “Supplemental Table 1” (lines: 134, 161, 173, 204, 213, 218, 324, 325, 338, 339,, 344, 365, 389, 424, 424, 469).

Renamed the “Materials” section to “Materials and methods” (line 101).

Removed use of “&” and replaced it with “and”.

The file names for all figures have been adjusted to align with in-text citation names, eg. Fig1.tif, Fig2.tif, etc.

Affiliations for authors were updated to reflect current positions (line 14).

We believe these revisions bring the paper into full compliance with PLOS ONE’s style requirements.

The Academic Editor wrote (Comment #2): “In your Methods section, please provide additional information regarding the permits you obtained for the work. Please ensure you have included the full name of the authority that approved the field site access and, if no permits were required, a brief statement explaining why.”

Response to Comment #2: We have added clarifying language to the Methods and materials section regarding the permissions granted for access to the hives and highlighting that permits were not required for the pollen collection since the hives are located on a University of Utah campus (lines: 117-119).

The Academic Editor wrote (Comment #3): “When completing the data availability statement of the submission form, you indicated that you will make your data available on acceptance. We strongly recommend all authors decide on a data sharing plan before acceptance, as the process can be lengthy and hold up publication timelines. Please note that, though access restrictions are acceptable now, your entire data will need to be made freely accessible if your manuscript is accepted for publication. This policy applies to all data except where public deposition would breach compliance with the protocol approved by your research ethics board. If you are unable to adhere to our open data policy, please kindly revise your statement to explain your reasoning and we will seek the editor's input on an exemption. Please be assured that, once you have provided your new statement, the assessment of your exemption will not hold up the peer review process.”

Response to Comment #3: We deeply appreciate PLOS ONE’s commitment to ensuring open data and in order to comply with the policy, we have:

Submitted broad and clean reads to the NCBI public repository

SubmissionID: SUB15836198

BioProject ID: PRJNA1381099

R scripts and appropriate files have been added to DRYAD

DOI: 10.5061/dryad.xksn02vw8

Upon completion of these tasks all relevant data from this paper is publicly available.

The Academic Editor wrote (Comment #4): “Please amend your authorship list in your manuscript file to include author Claudia Wiese.”

The Academic Editor wrote (Comment #5): “Please amend the manuscript submission data (via Edit Submission) to include author Claudia L. Wiese.”

Response to Comment #4 and #5: The authorship and the manuscript submissions have been amended to be consistent to reflect the author as “Claudia L. Wiese”.

The Academic Editor wrote (Comment #6): “Note that Figure 1 in your submission contain copyrighted images … We require you to either (1) present written permission from the copyright holder to publish these figures specifically under the CC BY 4.0 license, or (2) remove the figures from your submission”

Response to Comment #6: We have replaced the copyrighted image in Figure 1 with a non-copyrighted image.

Reviewer #1 wrote (Comment #1): “The sample size (n=6 hives) limits statistical inference. This should be clearly stated as a limitation in both Abstract and Discussion”

Response to Comment #1 : This explicit language has been added to both the Abstract (line: 8) and the Discussion (lines: 409 and 471-472).

Reviewer #1 wrote (Comment #2): “The PERMANOVA detected significant differences between treatments, but post-hoc comparisons were nonsignificant, suggesting a broad rather than treatment-specific effect. This nuance should be discussed more explicitly.”

Response to Comment #2: We have expanded our discussion of this nuance more thoroughly in the Discussion section (lines: 429-431). Specifically, we clarify that the non-significant post hoc results do not negate the significant PERMANOVA but it does indicate broader differences among samples and calls for further research.

Reviewer #1 wrote (Comment #3): “Clarify whether biological replicates (hives) or technical replicates (DNA extractions) formed the basis of statistical analysis.”

Response to Comment #3: It has been explicitly clarified in the Statistical methods subsection that our statistical analysis was based on two variants of biological replications (lines: 239-242). That is, we treated each hive as a replicate and, when possible, multiple 1gram subsamples from the same pollen sample were sequenced. We did not perform any technical replicates, for example by sequencing the exact same sample more than once.

Reviewer #1 wrote (Comment #4): “Adding effect size estimates (R² or beta-dispersion) and clarifying how variability was partitioned would strengthen conclusions.”

Response to Comment #4: A typo identified: R2 was mistakenly written as R2. This has been fixed (lines: 358 and 361).

Reviewer #1 wrote (Comment #5): “Dispersion differences (PERMDISP) were significant; this assumption violation should be explicitly discussed since it affects PERMANOVA interpretation.”

Response to Comment #5: We appreciate the reviewer’s note about the importance of this statistical violation. As a reminder, PERMDISP was non-significant for the analysis of treatment, however as noted it was significant for the analysis of location. We expanded discussion related to the significant PERMADISP in the Discussion section (lines: 466-472). However, it is important to note that NMDS plots (stress = 0.1224, where < 0.2 indicates a good fit) and diversity metrics are not affected by this dispersion difference.

Editor #1 wrote (Comment #6): “Diversity indices (Shannon, richness) are presented visually but could be supported by statistical comparison (e.g., ANOVA or Wilcoxon tests).”

Response to Comment #6:

To further quantitatively explore the diversity indices, we first tested whether the data was normally distributed, and since it was, we ran an ANOVA test. If the ANOVA test was significant, we then proceed with the post-hoc Tukey HSD analysis. We have included this analysis the following sections: Statistical methods, (line: 299-301), Results (lines: 398-402), and Discussion (lines: 431-434). We have also added a supplementary table (S12 Table) with significant results. We also added specific numbers to the figure caption of the related figure to allow further explicit quantification (line: 324-325). We want to note that supplementary tables (S10 and S11) include mean, median, standard deviation, minimum, maximum, and IQ values for both the Shannon and Richness diversity plots available to the reader.

Reviewer #1 wrote (Comment #7): “Include the R² values for treatment effects in Results and the Supplementary Tables for transparency.”

Response to Comment #7: The R2 values have now been added to the supplementary tables (S8 and S9 Tables, in supporting_information.docx). As answered above in Comment #4, a typo from R2 to R2 was changed in the Results section (lines: 358 and 361).

Reviewer #1 wrote (Comment #8): “The figures are scientifically appropriate and clearly convey the findings; however, several appear to be low-resolution raster images embedded in the Word manuscript. To meet publication quality, all figures should be exported in ≥300 dpi or, preferably, vector-based formats (PDF, EPS, or TIFF). Specific observations:

Figure 1 (photo + schematic): the photograph is slightly blurry, and fonts in the diagram are too small; scale or legend text should be enlarged.

Figures 2–3 (bar chart and heatmap): color palette uses similar hues, reducing contrast. Please use a colorblind-friendly palette and increase label font size.

Figure 4 (NMDS plots): acceptable but should be exported directly from R in vector format for clarity.

Figures 5–6 (diversity plots): scientifically correct but axis labels are too small; increase font and ensure legends remain legible at journal layout size.”

Response to Comment #8: We appreciate this feedback. We had no comments on figure quality from the Academic Editor and given some of the previous experiences from the authors on this paper, we are making the assumption that this is due to how the images were sent to you. To confirm, we have checked the formatting requirements by PLOS ONE to ensure font sizes are correct.

We have however changed the color palettes, used a color-blind friendly pallet for Figures 2 and 3. We did adjust Figure 1 as well.

Reviewer #1 wrote (Comment #9): “Highlight that hive-level effects may reflect genetic, environmental, or queen-related factors.”

Response to Comment #9: The understanding of hive-level effects and their impacts on foraging behaviors is not entirely clear in the current literature. While we understand there is substantive literature that notes that hives are different in their ability to be resistant to Varroa or that the queen can dictate aggressiveness of a hive, it is only assumed the hives forage differently as a result. Therefore we have added a line in the Discussion section that these factors should be considered in further research that parses apart factors that influence A. mellifera foraging choices (line: 473-477).

Reviewer #1 wrote (Comment #10): “Acknowledge that “hive identity” could also reflect temporal floral availability rather than intrinsic hive traits.”

Response to Comment #10: We have added language that clarifies our assumption that hives from the same location, assumed to physically have equal access to the same temporal resources, had slightly differing identities (line: 463-465). We also acknowledge that further research into hive foraging identities would have to parse apart temporal floral availability with the actual hive’s preferences (line: 478). Lastly, we have changed our use of “hive identity” to a “hive foraging identity" (line: 475, 478, 484) to further clarify that while hives have identities based on other factors, we were observing an identity as it relates to foraging choices. In this section about “hive foraging identity” we hope to only introduce a concept that was identified in our research, acknowledging that many factors are likely influencing a hive’s choice of where to forage.

Reviewer #1 wrote (Comment #11): Temper statements implying causation: “instead of ‘FA treatment has an impact,’ use ‘FA treatment was associated with detectable shifts in foraging composition.’”

Response to Comment #11: This comment is helpful, as we recognize that research on such a dynamic organism (often regarded as a “superorganism”) as A. mellifera, is rarely able to discern direct cause and effect, however it does not discount detected differences as irrelevant. We revised the Abstract (line:15) and Discussion (lines: 412-413, 441) to further attempt to properly temper statements that imply causation without entirely discounting the validity of the findings as they connect to the changes in foraging.

Reviewer #2 wrote: “The introduction has a solid theoretical fundamentation, the method is replicable and well detailed, the material is well discussed, and the conclusion aligns with the objectives. The references are also relevant to the research.”

Response to Reviewer 2: These comments do not indicate that any revisions need to be conducted.

---

## [Editor Report · Decision Letter 1]

12 Feb 2026

Effect of formic acid treatment on *Apis mellifera* foraging behavior using nanopore metabarcoding technologies

PONE-D-25-53512R1

Dear Dr. Wiese,

We’re pleased to inform you that your manuscript has been judged scientifically suitable for publication and will be formally accepted for publication once it meets all outstanding technical requirements.

Kind regards,

Kai Wang

Academic Editor

PLOS One
---

## [Editor Report · Acceptance letter]

PONE-D-25-53512R1

PLOS One

Dear Dr. Wiese,

I'm pleased to inform you that your manuscript has been deemed suitable for publication in PLOS One. Congratulations! Your manuscript is now being handed over to our production team.

Kind regards,

on behalf of

Dr. Kai Wang

Academic Editor

PLOS One